# Development of a DNA Metabarcoding Method for the Identification of Insects in Food

**DOI:** 10.3390/foods12051086

**Published:** 2023-03-03

**Authors:** Sophie Hillinger, Julia Saeckler, Konrad J. Domig, Stefanie Dobrovolny, Rupert Hochegger

**Affiliations:** 1Institute of Food Science, Department of Food Science and Technology, University of Natural Resources and Life Sciences, Vienna, Muthgasse 18, 1190 Vienna, Austria; 2Department for Molecular Biology and Microbiology, Institute for Food Safety Vienna, Austrian Agency for Health and Food Safety, Spargelfeldstrasse 191, 1220 Vienna, Austria

**Keywords:** insects, DNA metabarcoding, food authenticity, species identification, NGS

## Abstract

Insects have the potential to become an efficient and reliable food source for humans in the future and could contribute to solving problems with the current food chain. Analytical methods to verify the authenticity of foods are essential for consumer acceptance. We present a DNA metabarcoding method that enables the identification and differentiation of insects in food. The method, developed on Illumina platforms, is targeting a 200 bp mitochondrial 16S rDNA fragment, which we found to be suitable for distinguishing more than 1000 insect species. We designed a novel universal primer pair for a singleplex PCR assay. Individual DNA extracts from reference samples, DNA extracts from model foods and food products commercially available were investigated. In all of the samples investigated, the insect species were correctly identified. The developed DNA metabarcoding method has a high potential to identify and differentiate insect DNA in the context of food authentication in routine analysis.

## 1. Introduction

Limited land available for livestock production, increasing greenhouse gas emissions and a rising world population are becoming a challenge for the growing meat and protein demand worldwide [1]. Insects have the potential to become a sustainable, efficient, and reliable source of food for humans and overcome some of the burdens of the meat industry [2]. Depending on their species and state of metamorphosis, insects can contain remarkable amounts of proteins, calories, fat, vitamins, and minerals, and therefore complement or even replace meat in the human diet [3]. In large parts of the world such as Africa, Asia and South America, the so-called entomophagy (the consumption of insects as a food source for humans) is common in traditional cuisine [1]. Insects are widespread in many regions worldwide and comparatively easy to propagate. Since many years, they have been among the most important sources of nutrients, especially for developing countries that are regularly affected by starvation [3,4]. In Europe, entomophagy is not yet widespread, but since the 21st century, public and economic interest has been growing due to EU subsidies.

The production and placing on the market of insects and parts thereof are regulated in Europe by the legislation on Novel Foods [5]. The yellow mealworm (*Tenebrio molitor*), rather their larvae, was the first insect to be approved in the EU, followed by the migratory locust (*Locusta migratoria*), the house cricket (*Acheta domesticus*), and the buffalo worm larvae (*Alphitobius diaperinus*) [6,7,8,9]. Further insect species can be expected to be approved by the EU. Insects can not only be an intentional component in food products but can also occur unintentionally as storage pests [3]. Due to possible production of allergenic substances and symbiosis with mycotoxins, the presence of pests can be of great importance to human health [10]. Food authenticity is important in terms of food fraud, the quality and safety of ingredients and cross-contamination. Food can be considered authentic if it is in its original state and complies with its declaration. Premium or high-priced products are especially prone to be adulterated by cheap or low-quality ingredients and thus need to be verified by analytical methods to support control [11]. At least the insects recently approved for food use should be detectable and discriminable from other insect species. DNA-based analytical methods are gaining more and more importance because they enable specific and fast analysis and have a broad range of applications. DNA is not only present in almost all foods but is also quite heat tolerant and can therefore be used as a parameter in processed foods [12]. Polymerase chain reaction (PCR) assays for insect identification have been developed in singleplex and multiplex PCR format [13,14]. However, the number of detectable insects is low due to the currently still small number of detection methods for insects in food. Multiplex methods are also limited in the number of optical channels in the detection unit of the real-time PCR device. In addition, a method should allow the analysis of highly processed foods, but published assays may fail to amplify degraded DNA because the designed primer system forms PCR products that are too long. These limitations can be overcome by using methods of barcode sequencing with universal primer systems. DNA barcodes are usually composed of conserved regions at both ends and a variable part between the primer binding sites to discriminate between the species of interest [15,16]. In traditional DNA barcoding, PCR products gained through amplification of the designated DNA barcode region, e.g., cytochrome oxidase I gene [17] are then subjected to Sanger sequencing [18,19,20]. To increase the efficiency of this method, a combination of DNA barcoding with next generation sequencing (NGS) is favorable [21,22]. So-called DNA metabarcoding enables the detection of a larger number of species simultaneously and identifying them through reference sequences [23,24]. For this purpose (correct) database entries are required [25]. DNA metabarcoding methods to identify and differentiate species have already been developed and published, e.g., the detection of mammals and birds in foods, and bivalve species in seafood [26,27,28,29].

In this study, we aimed to develop a DNA metabarcoding method that uses relatively short PCR products of approximately 200 bp in length, allowing identification and differentiation of insect species in processed food products. The method was developed using the Illumina MiSeq^®^ and iSeq^®^ platforms.

## 2. Materials and Methods

### 2.1. Samples

Insect samples (pure material of individual species) from the Institute for Sustainable Plant Production, Vienna, Austrian Agency for Health and Food Safety (AGES) and insect-containing food obtained from supermarkets and online shops were used for the experiments. Experts at the Institute for Sustainable Plant Production confirmed identity of the insect species used as reference samples. Preferably, reference samples were ordered alive or alternatively already dried or frozen. Furthermore, self-made insect cookies and burgers (which have served as model foods in previous studies) were obtained from the Food Control Authority of the Canton of Zurich, Zurich, Switzerland. The cookies contained three insect species in equal proportions, while the burgers had an asynchronous composition from 0.1 to 10% [14]. All samples were kept at a temperature of −20 °C until DNA extraction was performed. The samples used for the development of the method, including the four insect species commonly consumed in Europe, are listed in Table 1. The selection criterion was the affiliation of these insects to the main representatives of edible insects [1].

### 2.2. DNA Extraction

At the beginning, all samples were either cut into smaller pieces or homogenized in a mortar or lab mill. After that step, samples were lysed in the presence of a hexadecyltrimethylammonium bromide/polyvinylpyrrolidone extraction solution (CTAB/PVP-buffer) and proteinase K at elevated temperature under constant shaking. Then, DNA extraction was performed using a commercially available kit. The Maxwell RSC PureFood GMO and Authentication Kit from Promega (Madison, WI, USA) and the Maxwell^®^ 16 instrument (Promega, Madison, WI, USA) were used for DNA isolation following the manufacturer’s instructions. The DNA extraction procedure was verified by including negative and positive extraction controls. The yield of the DNA extracts was measured fluorometrically with the fluorometer using Qubit^®^ 2.0 fluorometer (Thermo Fisher Scientific, Waltham, MA, USA). The Qubit^®^ dsDNA broad range assay kit (2–1000 ng) and, for low DNA concentrations, the Qubit^®^ dsDNA high sensitivity assay kit (0.2–100 ng) were used according to the manufacturer’s protocol. The purity of the DNA was also checked using the ratio of the absorption at 260 and 280 nm (QIAxpert spectrophotometer, Qiagen, Hilden, Germany). The DNA extracts were frozen at −20 °C until further use.

### 2.3. Reference Sequences and Primer Design

We used the “Worldwide list of recorded edible insects, Jongema, 2017” as the basis for our search for insect DNA sequences. Reference sequences in FASTA format for individual species were downloaded from the National Center for Biotechnology Information (NCBI, Bethesda, MD, USA) and imported into the CLC Genomics Workbench software (version 11, Qiagen, Hilden, Germany). Preferably, entire mitochondrial DNA sequences were derived from the NCBI RefSeq database due to their expert-proven reliability. The sequences of the mitochondrial 16S rDNA were extracted from the complete genomes and multiply aligned by using the default settings of the CLC Genomics Workbench software (version 11, Qiagen, Hilden, Germany). The primers used were manually designed for this multiple sequence alignment. Four forward and three revers primers have been designed and tested in 12 combinations to amplify a ~200 bp barcode region of mitochondrial 16S ribosomal DNA from different insect species. The sequences of the primers tested are shown in Table 2. The formation of primer dimers was checked by using the OligoAnalyzer Tool provided by Integrated DNA Technologies (IDT, Coralville, IA, USA). Calculations of the annealing temperature of the primers were performed using specialized computer programs as displayed in the TIB Molbiol product description (Berlin, Germany). The target-specific primers, including the overhang adapter sequences were purchased from TIB Molbiol (Berlin, Germany).

To verify the successful amplification of the designed primers, real-time PCR of DNA from positive controls was performed (PCR results of the reference samples are shown in Appendix A (Appendix A)). The individual DNA extracts of the insect species were used as reference samples or positive controls (see Table 1). During PCR-optimization, the DNA input amount of 12.5 ng and the amount of ‘ready-to-use’ HotStarTaq Master Mix Kit from Qiagen (Hilden, Germany) were kept constant and applied as previously published [26]. The annealing temperature (58–62 °C), primer concentrations (final concentrations 0.1–0.8 µM), the addition of magnesium chloride solution (1.5 mM or 3 mM MgCl_2_) and PCR cycle numbers (30, 35 and 40) were varied. Real-time PCR reactions were carried out using a fluorescent intercalating dye (EvaGreen^®^ (20× in water)) in 96-well plates on the LightCycler^®^ 480 System (Roche, Penzberg, Germany). The correct length of the PCR products was checked by agarose gel electrophoresis, and melting curve analysis was used to detect any non-specific artifacts. The volume of the PCR reactions was 25 µL, made up of 22.5 µL reaction mix and 2.5 µL of diluted DNA sample (5 ng/µL) as template. For the no-template control (NTC), water was used instead of DNA. Possible contamination is checked by including negative extraction controls. The reaction mixture comprises Master Mix with fluorescent dye, primers, nuclease-free water and no or additional magnesium chloride solution.

### 2.4. Library Preparation and NGS

DNA sequencing of the samples was performed using the MiSeq^®^ and iSeq^®^ 100 platform from Illumina (San Diego, CA, USA). The DNA extracts were typically diluted to a concentration of 5 ng/μL, those with a lower concentration were used undiluted.

DNA libraries were prepared as described previously [26] with minor modifications (magnetic beads volume: 36 µL; average library size: 226 bp; the iSeq^®^ 100 instrument denatured the diluted libraries automatically during the sequencing process). The DNA library was diluted with 10 mM Tris-HCL at pH 8.6 to the concentration of 4 nM (MiSeq^®^) or 1 nM (iSeq^®^ 100), respectively. The concentration of the pooled DNA libraries (5 µL for MiSeq^®^ and 7 µL for iSeq^®^ 100) was measured using the Qubit^®^ 4.0 fluorometer (Thermo Fisher Scientific, Waltham, MA, USA). All paired-end sequencing runs were carried out using either the iSeq^®^ 100 i1 Reagent v2 (300 cycles) or MiSeq^®^ Reagent Kit v2 (300 cycles) at a final loading concentration of 8 pM. A 5% PhiX spike-in was used as sequencing control.

Reference samples and the DNA extracts from model foods were sequenced on both sequencing platforms (two sequencing runs, one replicate per run). The commercial food products were sequenced with the MiSeq^®^ or the iSeq^®^ 100 platform (Illumina, San Diego, CA, USA). The obtained DNA sequences of the reference samples were compared after sequencing on the MiSeq^®^ and the iSeq^®^ 100 platform for each individual reference sample. The data of the sequence comparison are presented in Appendix A.

### 2.5. NGS Data Analysis Using Galaxy

The raw sequence image data (bcl-files) was processed with the help of the instrument’s conversion software bcl2fastq2 (version 2.19.0.316, Illumina, San Diego, CA, USA) and DNA FastQ files were generated. The resulting FastQ files were used as input for bioinformatics tools performing downstream analysis. For downstream analysis, the published and adapted analysis pipeline in Galaxy (version 19.01) was used [29]. Taxonomic assignments were made by aligning dereplicated sequences against a customized database of pre-assigned reference sequences (DNA barcodes) provided by NCBI using BLASTn [30]. The entries of the current AGES customized database are listed in Appendix A. The AGES customized database contained entries from insects assigned to the nine orders *Blattodea* (cockroaches), *Isoptera* (termites), *Coleoptera* (beetles), *Diptera* (flies), *Hemiptera* (cicadas, bugs; suborders: *Coleorrhyncha*, *Heteroptera*, *Sternorrhyncha*), *Hymenoptera* (wasps, bees, and ants), *Lepidoptera* (butterflies and moths), *Odonata* (dragonflies and damselflies), and *Orthoptera* (grasshoppers, locusts, and crickets). Furthermore, included: *Archaeognatha*, *Zygentoma*, *Ephemeroptera*, *Plecoptera*, *Embioptera*, *Notoptera*, *Dermaptera*, *Mantodea*, *Phasmatodea*, *Mantophasmatodea*, *Zoraptera*, *Psocoptera*, *Phthiraptera*, *Thysanoptera*, *Raphidioptera*, *Megaloptera*, *Neuroptera*, *Strepsiptera*, *Trichoptera*, *Mecoptera*, *Siphonaptera*, *Cicadomorpha*, *Fulgoromorpha*, and *Arachnida*. In addition, a simultaneous comparison of all DNA sequences was performed with CLC Genomics Workbench (version 11, Qiagen, Hilden, Germany).

## 3. Results and Discussion

The purpose of the present study was to develop a DNA metabarcoding method, which can be used for the authentication of various insect species and products thereof. The samples tested consisted of 18 references samples, DNA extracts from insect cookies and burgers, as well as from commercial food products.

Therefore, we focused our search on DNA barcodes no longer than 200 bp to enable detection of species in raw and processed insect-containing food products. Mitochondrial DNA, in particular the mitochondrial 16S ribosomal DNA gene, was chosen as a source of markers since we have already used this gene for our mammalian and poultry assay. Furthermore, the DNA libraries should be sequenced with 300-cycle Illumina reagent kits to allow for the simultaneous analysis of insect samples along with those of mammalian and poultry species using the recently published DNA metabarcoding method [26]. We designed primers targeting a region of the mitochondrial 16S ribosomal RNA gene (Table 2). All primers were tested for their applicability on the different DNA extracts from reference samples (Table 1). With the primer pair Fwd-I-3 and Rev-I-1 a high amount of PCR products with the expected length was obtained and thus, this primer set was considered applicable for use in practice. The optimal PCR-conditions were determined as follows: HotStar-Taq Master Mix Kit (Qiagen, Hilden, Germany) was used in the reaction mix at a final concentration of 1x and the final concentration of primers was 0.4 µM in the presence of additional magnesium chloride (final concentration of magnesium 4.5 mM). The PCR protocol involved a 15 min initial denaturation at 95 °C, followed by 35 cycles of 30 s each at 95 °C, 58 °C, and 72 °C, and a 10 min final elongation at 72 °C. An alignment of selected DNA barcode sequences for relevant insect species is displayed in Figure 1. The binding positions of forward and reverse primer are marked in blue and green, respectively.

The pairwise comparison tool of the CLC Genomics Workbench software was used to compare the selected DNA barcode region of 1100 insect species to identify similarities and differences. A typical graphical representation of a sequence comparison of DNA sequences from the 18 reference samples is shown in Figure 2. A color scheme is used to highlight the relationship between the DNA barcode regions, with blue representing differences and dark red representing high similarity in the variable region of the DNA sequences. Analysis of the data showed that 92% of all insects (DNA barcodes) under investigation can be discriminated from each other. The sequence alignment data revealed that the selected DNA barcode region cannot discriminated between all species of the following genera: *Drosophila* spp., *Chrysomya* spp., *Bactrocera* spp., *Cheumatopsyche* spp., *Sinopodisma* spp., *Fruhstorferiola* spp., *Chorthippus* spp., *Stenocatantops* spp., *Gomphocerus* spp., *Traulia* spp., *Filchnerella* spp., *Bryodema* spp., *Oedaleus* spp., *Tetrix* spp., *Cryptolestes* spp., *Anax* spp., *Euphaea* spp., *Actias* spp., *Dendrolimus* spp., *Ostrinia* spp., *Magicicada* spp., *Bombus* spp., *Bryodemella* spp., *Culex* spp., *Pomacea* spp., *Pontia* spp., *Rapisma* spp., *Traulia* spp. and *Vespa* spp. Furthermore, the following pairings cannot be distinguished with the developed marker system, because the base sequence in the variable region of the amplified barcode is identical: *Musca domestica*:*Dasyhippus barbipes*, *Pliacanthopus bimaculatus*:*Miromantis yunnanensis*, *Omocestus viridulus*:*Dnopherula yuanmowensis*, *Pseudoeoscyllina brevipennisoides*:*Euchorthippus unicolor*, *Pseudotmethis rubimarginis*:*Filchnerella rubrimargina*:*Filchnerella helanshanensis*, *Filchnerella qilianshanensis*:*Sinotmethis brachypterus*, *Bryodema kozlovi*:*Bryodema nigroptera*:*Bryodemacris uvarovi*:*Bryodemella tuberculata diluta*, *Bryodemella holdereri holdereri*:*Bryodema dolichoptera*:*Angaracris rhodopa*, *Oedaleus abruptus*:*Parapleurus alliaceus*.

### 3.1. Analysis of DNA Extracts from Reference Samples

The optimized DNA metabarcoding method was applied to identify insect species in individual DNA extracts from reference samples. The results obtained by DNA metabarcoding are shown in Table 3. The table displays average values for the total raw reads, the total reads that passed the analysis pipeline in Galaxy, and the total reads correctly assigned to the eighteen species (based on two replicates, with one exception for “*Pachnoda marginata*”). The number of correctly assigned reads ranged from approximately 17,000 and 148,000 reads for the selected number of samples for the sequencing experiment, resulting in a clear identification of the insect species. The four EU-approved edible insect species and all other insect species tested were identified at a high rate (>97% identity with reference sequences) using this workflow. The comparison of the DNA sequences after sequencing of the reference samples on the MiSeq^®^ and the iSeq^®^ 100 platform showed no deviation (Appendix A). Although all of the eighteen reference samples were correctly assigned on both sequencing platforms, in case of *Galleria mellonella* (Greater wax moth), *Gryllodes sigillatus* (Tropical house cricket), *Plodia interpunctella* (Indian meal moth), and *Lethocerus indicus* (Water bug) the obtained sequences by next generation sequencing were not identical to the reference sequences. There were up to four mismatches between the reads of the individual representative sequences and the corresponding reference sequences in the user-defined database imported from NCBI, indicating gaps or errors in the database (Appendix A).

### 3.2. Analysis of DNA Extracts from Model Foods

We investigated the suitability of the DNA metabarcoding method for processed and heat-treated food samples with known insect species composition. Therefore, we analyzed model food products (five insect cookies and four insect burgers) from a Swiss laboratory from a previous research project. A detailed product information is given in Köppel et al., 2019 [14]. In general, the cookies and burgers contained three insect species (*Tenebrio molitor*, *Acheta domesticus*, *Locusta migratoria*) in a ratio from 0.1 to 10.0% (*w*/*w*) in the presence of wheat flour or ground meat, respectively. The results obtained for the nine model foods are summarized in Table 4. The DNA metabarcoding method allowed the correct and sensitive identification of all insect species present down to a spiking level of 0.1% in model food samples. It was also shown that the barcode developed, with a length of 200 base pairs, allows discrimination of the three insect species in the heat-treated model samples, even if the spiking material was prepared asynchronously. These results indicate that the DNA metabarcoding method based on the primer set Fwd-3 and Rev-I-1 is applicable for the detection of insect species in processed food products.

### 3.3. Analysis of Insect Samples Commercially Available

To assess the suitability of our DNA metabarcoding method to commercially available foods, 38 food products declared to contain insects were analyzed. According to the declaration, 23 samples (1–23) should contain buffalo worm species, four samples (24–27) should contain mealworm species, and 11 samples (28–38) should contain cricket species. In order to represent a wide spectrum of available products, both pure insect samples in dried, milled or roasted form and mixed products with a very low insect content of only 0.1% were selected. The results showed that insect DNA could be detected in all samples, and the number of correctly assigned sequences reached at least about 73,000 reads. Table 5 summarizes the results obtained for the 38 commercial food products from supermarkets and online stores. Our results confirmed the presence of the three species according to their declaration and the suitability of the method for the identification of insect components down to a presence of only 0.1%.

## 4. Conclusions

DNA metabarcoding is considered an advanced tool for monitoring food authenticity, a reference method for the detection of animal species and birds has already been developed [29]. In this study we developed a DNA metabarcoding method that has great potential for identifying insect species in food and can serve as an effective screening method for species authentication in food products that may contain insects. A singleplex PCR assay was developed for the amplification of the short DNA target region of the mitochondrial 16S rDNA gene that serves as a DNA barcode. The applicability of the novel DNA metabarcoding method was investigated by analyzing individual DNA extracts from reference samples, nine heat-treated model foods, as well as DNA extracts from 38 commercially available food products. Analysis of the tested samples demonstrated that the method is suitable for insect identification, even in processed or complex foods down to an insect content of only 0.1%. This sensitivity was also achieved in model foods with asynchronous composition of insect-containing ingredients. There were 38 commercial foods with declared insect ingredients, including compound products and pure products in dried, roasted and powdered form, that were checked for correct labeling. Noticeably, the declared insect ingredient was confirmed in all 38 commercial products tested.

For many insects, PCR methods are lacking for reliable detection, so a major advantage of DNA metabarcoding is the simultaneous detection of a large number of insect species in one testing approach. To determine further performance parameters of the DNA metabarcoding method presented and to assess whether the method also allows semi-quantitative statements, an interlaboratory validation of the method should be carried out. Although the insect species currently relevant in food production were clearly detected, successful differentiation on species level was not possible for all samples examined in silico.

A limiting factor for the application of metabarcoding methods is still the lack of sequencing equipment in laboratories and gaps in sequence database content, especially for insect species. The transferability of the method to different platforms, it runs successfully on both Illumina MiSeq^®^ and iSeq^®^ 100 instruments, and by combining different applications (joint sequencing of plant and animal species, bacteria, etc.), the costs can be kept sufficiently low, should laboratories consider purchasing such equipment.

## Figures and Tables

**Figure 1 foods-12-01086-f001:**
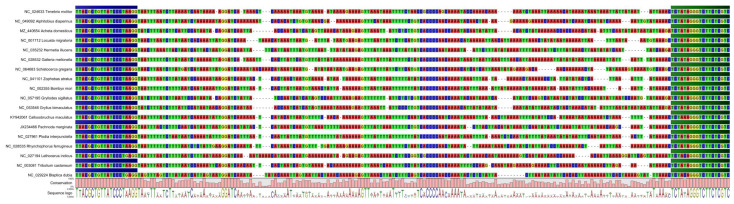
Multi-species sequence alignment of the mitochondrial 16S rDNA barcoding region of edible and relevant insects in Europe. The universal primer set binding sites are highlighted by blue (forward primer) and green (reverse primer) colored bars (CLC Genomics Workbench 11, Qiagen, Hilden, Germany).

**Figure 2 foods-12-01086-f002:**
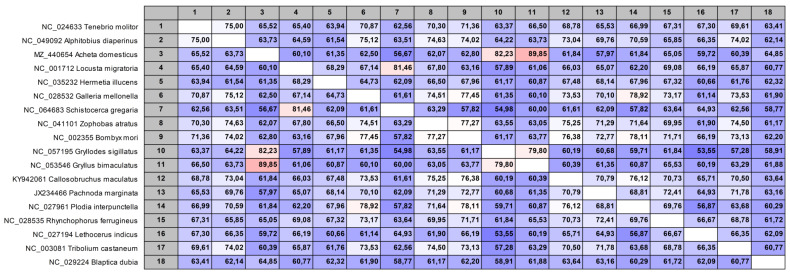
Multiple alignment of 18 reference samples. The figure was created with the pairwise comparison tool (CLC Genomics Workbench 11, Qiagen, Hilden, Germany).

**Table 1 foods-12-01086-t001:** Insect species used to develop the DNA metabarcoding method.

Scientific Name	Commercial Name (English)
*Tenebrio molitor*	Mealworm
*Alphitobius diaperinus*	Buffalo worm
*Acheta domesticus*	House cricket
*Locusta migratoria*	Migratory locust
*Hermetia illucens*	Black soldier fly
*Galleria mellonella*	Greater wax moth
*Schistocerca gregaria*	Desert locust
*Zophobas atratus*	Superworm
*Bombyx mori*	Silkworm moth
*Gryllodes sigillatus*	Tropical house cricket
*Gryllus bimaculatus*	Mediterranean field cricket
*Callosobruchus maculatus*	Cowpea weevil
*Pachnoda marginata*	Sun beetle
*Plodia interpunctella*	Indian meal moth
*Rhynchophorus ferrugineus*	Red palm weevil
*Lethocerus indicus*	Water bug
*Tribolium castaneum*	Red flour beetle
*Blaptica dubia*	Cockroach

**Table 2 foods-12-01086-t002:** Primer sequences tested in this study.

Name	Sequence 5′ → 3′
Fwd-I-1	TTACGCTGTTATCCCTAAGG
Fwd-I-2	TAACGCTGTTATCCCTAAGG
Fwd-I-3	TWACGCTGTTATCCCTAAGG
Fwd-I-4	ACGCTGTTATCCCTAAGG
Rev-I-1	GACGAGAAGACCCTATAGA
Rev-I-2	GACGATAAGACCCTATAGA
Rev-I-3	GACGAKAAGACCCTATAGA
**Illumina Overhang Adapter Sequences**	
Forward	TCGTCGGCAGCGTCAGATGTGTATAAGAGACAG
Reverse	GTCTCGTGGGCTCGGAGATGTGTATAAGAGACAG

**Table 3 foods-12-01086-t003:** Results for DNA extracts from reference samples. Numbers are average values (*n* = 2, two sequencing runs, one replicate/run).

Scientific Name	Identified Species	Total Raw Reads	Total Reads Passing the Pipeline	Reads Assigned Correctly
*Tenebrio molitor*	*Tenebrio molitor*	99,977	85,871	85,681
*Alphitobius diaperinus*	*Alphitobius diaperinus*	141,850	124,433	124,190
*Acheta domesticus*	*Acheta domesticus*	131,959	114,260	114,137
*Locusta migratoria*	*Locusta migratoria*	98,907	88,917	87,041
*Hermetia illucens*	*Hermetia illucens*	57,123	53,160	53,003
*Galleria mellonella*	*Galleria mellonella*	55,899	36,539	36,361
*Schistocerca gregaria*	*Schistocerca gregaria*	126,910	109,280	109,168
*Zophobas atratus*	*Zophobas atratus*	106,749	102,126	102,051
*Bombyx mori*	*Bombyx mori*	57,587	56,222	56,158
*Gryllodes sigillatus*	*Gryllodes sigillatus*	145,814	115,026	114,938
*Gryllus bimaculatus*	*Gryllus bimaculatus*	79,639	68,255	68,196
*Callosobruchus maculatus*	*Callosobruchus maculatus*	51,577	50,403	50,352
*Pachnoda marginata*	*Pachnoda marginata*	17,830	17,427	17,414
*Plodia interpunctella*	*Plodia interpunctella*	70,974	64,860	64,728
*Rhynchophorus ferrugineus*	*Rhynchophorus ferrugineus*	98,335	93,398	93,289
*Lethocerus indicus*	*Lethocerus indicus*	97,715	89,487	89,301
*Tribolium castaneum*	*Tribolium castaneum*	160,925	148,201	148,040
*Blaptica dubia*	*Blaptica dubia*	151,001	115,351	115,293

**Table 4 foods-12-01086-t004:** Results obtained from model foods representing three insects of interest. Numbers are average values (*n* = 2, two sequencing runs, one replicate/run).

	Model Foods		Total Number of Raw Reads	Total Number of Reads Passing the Pipeline	Reads Assigned Correctly		
**Cookies (Main component: wheat flour)**					**Species 1**	**Species 2**	**Species 3**
**Insect cookie 1 (proportion insects: 10% (*w*/*w*))**			110,506	105,462			
*Locusta migratoria*	*Acheta domesticus*	*Tenebrio molitor*			11,521	79,493	14,419
**Insect cookie 2 (proportion insects: 3.2% (*w*/*w*))**			113,085	108,271			
*Locusta migratoria*	*Acheta domesticus*	*Tenebrio molitor*			16,578	82,318	9259
**Insect cookie 3 (proportion insects: 1% (*w*/*w*))**			103,454	98,752			
*Locusta migratoria*	*Acheta domesticus*	*Tenebrio molitor*			22,975	65,476	10,247
**Insect cookie 4 (proportion insects: 0.32% (*w*/*w*))**			76,268	72,307			
*Locusta migratoria*	*Acheta domesticus*	*Tenebrio molitor*			17,810	45,327	9112
**Insect cookie 5 (proportion insects: 0.10% (*w*/*w*))**			72,699	68,225			
*Locusta migratoria*	*Acheta domesticus*	*Tenebrio molitor*			13,204	43,988	10,993
**Burgers (Main component: ground meat, asynchronous preparation)**							
**Insect burger 1 (proportion insects: 3.2% (*w*/*w*))**			126,563	110,249			
*Locusta migratoria*	*Acheta domesticus*	*Tenebrio molitor*			1104	97,062	12,030
**Insect burger 2 (proportion insects: 1% (*w*/*w*))**			98,628	60,013			
*Locusta migratoria*	*Acheta domesticus*	*Tenebrio molitor*			45,442	12,137	2387
**Insect burger 3 (proportion insects: 0.32% (*w*/*w*))**			97,647	82,260			
*Locusta migratoria*	*Acheta domesticus*	*Tenebrio molitor*			63,118	18,658	387
**Insect burger 4 (proportion insects: 0.1% (*w*/*w*))**			111,195	100,029			
*Locusta migratoria*	*Acheta domesticus*	*Tenebrio molitor*			5080	94,180	744

**Table 5 foods-12-01086-t005:** Results obtained from commercial insect samples. Samples were sequenced either with the MiSeq^®^ or the iSeq^®^ 100.

Sample ID	Food Product	Insect Species Labeled (Amount in %, if Available)	Identified Species	Total Number of Raw Reads	Total Number of Reads Passing the Pipeline	Reads Assigned Correctly
1	Insect mushroom soup	Buffalo worm meal (26%)	*Alphitobius diaperinus*	116,024	112,783	112,695
2	Insect pancake ready mix	Buffalo worm meal (7.1%)	*Alphitobius diaperinus*	139,113	134,493	134,361
3	Insect beetroot risotto mix	*Alphitobius diaperinus*	*Alphitobius diaperinus*	132,667	128,963	128,839
4	Insect brownie mix	Buffalo worm meal (5.5%)	*Alphitobius diaperinus*	151,134	146,191	145,884
5	Insect oatmeal patty	*Alphitobius diaperinus*	*Alphitobius diaperinus*	100,524	97,449	97,409
6	Insect bread	Buffalo worm meal (4.8%)	*Alphitobius diaperinus*	106,194	102,849	102,635
7	Chocolate with worm	*Alphitobius diaperinus* (0.1%)	*Alphitobius diaperinus*	127,168	123,370	123,302
8	Insect meal (buffalo worms)	*Alphitobius diaperinus* (100%)	*Alphitobius diaperinus*	120,078	116,090	116,007
9	Insect bar sour cherry	Milled buffalo worms (12%)	*Alphitobius diaperinus*	155,677	150,368	150,267
10	Insect bar apple strudel	Milled buffalo worms (12%)	*Alphitobius diaperinus*	132,100	127,302	127,221
11	Insect bar apricot	Milled buffalo worms (13%)	*Alphitobius diaperinus*	175,897	170,155	170,112
12	Protein drink with cocoa taste	Buffalo worm protein (50%)	*Alphitobius diaperinus*	137,082	132,930	132,887
13	Protein drink with strawberry taste	Buffalo worm protein (50%)	*Alphitobius diaperinus*	117,959	114,836	114,761
14	Protein drink with vanilla taste	Buffalo worm protein (50%)	*Alphitobius diaperinus*	134,025	129,917	129,846
15	Protein drink with banana taste	Buffalo worm protein (50%)	*Alphitobius diaperinus*	112,744	109,573	109,482
16	Mealworm flour	Mealworm	*Alphitobius diaperinus*	115,509	111,621	111,530
17	Porridge with chocolate-hazelnut taste	Buffalo worms (20%)	*Alphitobius diaperinus*	129,371	125,431	125,395
18	Porridge with raspberry-coco taste	Buffalo worms (20%)	*Alphitobius diaperinus*	118,337	114,805	114,699
19	Insect peanut creme	Buffalo worm protein (17%)	*Alphitobius diaperinus*	126,108	122,465	122,420
20	Insect meal (buffalo worms)	Buffalo worm	*Alphitobius diaperinus*	128,800	125,408	125,348
21	Energy bar with roasted worms	Dried buffalo worms (10%)	*Alphitobius diaperinus*	135,324	130,434	130,348
22	Insect candy lollipop	Dried buffalo worms	*Alphitobius diaperinus*	215,121	209,388	209,339
23	Insect burger patty	*Alphitobius diaperinus* (38%)	*Alphitobius diaperinus*	104,859	101,862	101,861
24	Insect meal (mealworms)	*Tenebrio molitor* (100%)	*Tenebrio molitor*	87,354	73,377	73,242
25	Dark chocolate with mealworms	Roasted freeze-dried mealworms (2%)	*Tenebrio molitor*	104,484	89,683	89,535
26	Milk chocolate with mealworms	Roasted freeze-dried mealworms (2%)	*Tenebrio molitor*	92,543	76,521	76,371
27	Insect candy lollipop	Roasted freeze-dried mealworms	*Tenebrio molitor*	115,989	100,325	100,213
28	Crickets	Cricket	*Acheta domesticus*	105,620	102,960	102,931
29	Cricket flour	Cricket	*Acheta domesticus*	121,121	115,782	115,608
30	Cricket pasta	Cricket	*Acheta domesticus*	241,867	232,106	232,037
31	Fried crickets	Cricket	*Acheta domesticus*	99,982	97,911	97,854
32	Spicy crickets	*Acheta domesticus* (90%)	*Acheta domesticus*	103,818	98,562	98,519
33	Smoked crickets	Cricket	*Acheta domesticus*	110,766	107,170	107,050
34	Insect candy lollipop	Roasted freeze-dried crickets	*Acheta domesticus*	173,377	164,565	164,495
35	Cricket cracker	*Acheta domesticus* (15%)	*Acheta domesticus*	150,410	143,659	143,574
36	Cricket cracker	*Acheta domesticus* (15%)	*Acheta domesticus*	130,263	124,414	124,295
37	Cricket cracker	*Acheta domesticus* (15%)	*Acheta domesticus*	134,728	129,629	129,506
38	Insect bar	Cricket meal (15%)	*Acheta domesticus*	133,735	127,033	126,929

## Data Availability

The datasets generated during the current study are available from the corresponding author on reasonable request.

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
