# Peer review of "Development of a DNA Metabarcoding Method for the Identification of Insects in Food"

_foods, 2023, doi:10.3390/foods12051086_

Round 1
Reviewer 1 Report
In this study, the authors developed a DNA metabarcoding method for the identification of insects in food, by sequencing a fragment of 16S rRNA mitochondrial gene.
They highlighted that the designed primers were able to identify and differentiate insects from food.
The topic is extremely relevant for the journal and I really enjoyed reading this manuscript. However, I have major concern with this manuscript in terms of the organization of the manuscript and about NGS Methods.
In particular, the introduction section might to be improved by better contextualizing the research problem and the problem statement.
Similarly, materials and methods section is not well organized. For instance, the sampling is not clear (what means that you sampled different types of samples?). It was possible to get this information only in the results section. Also, why did you select that insect species? Please, define the criterion used. In DNA extraction, why did you use two instruments to quantify the DNA? In primers design, why did you analyzed PCR products with melting curves and / or agarose gel?
In library preparation, why the sequencing was performed by two instruments? In my opinion it should be better if you performed the experiment using always the same instrument in the same run.
Also, the metabarcoding procedures must include also negative controls (the author did not declared the use of negative controls when extracting DNA nor in the NGS procedure) and it must include replication.
In results, it should be useful to provide to readers supplementary data regarding metabarcoding results. In commercial samples, did you detect only declared species ? Did you deposited metabarcoding data?
Moreover, in this paper there is little discussion of results, indeed in the discussion section the authors do not really discuss their results, nor do they compare them and/or support them with data from other studies.
Similarly, conclusions need to be improved. For instance, I think that the success of the results did not depend only by the instrument used.
Reviewer 2 Report
The work is well presented and highlights a very interesting analysis that can be very useful in food industry considering the growing interest of insects as food ingredients. The experimental design as well as the results and discussion are clear and well presented.
Reviewer 3 Report
The authors present a DNA metabarcoding method that enables the identification and differentiation of insects in food. Insect food has been attracting attention in recent years as one means of solving the food problem, and it is thought that the means of quality assurance as food will become important in the future.
The experimental design is well organized and the results are clearly presented. However, since PCR for 16S rDNA is greatly affected by contamination, I think that indicating the results of real-time PCR in 2.3, including NTC, as supplementary data will help the reader's understanding.
In order for the reader to understand this manuscript more accurately, it is preferable to make the following modifications.
1, Supplementary data should include PCR results for individual samples (Mealworm, Buffalo worm, etc.) to demonstrate that the primers are working. You need to make sure that you don't get false positives for other samples when you use individual primers.
2, Please add a discussion about whether the model food results in table 4 match the manufacturer's published percentages of insects, to explain whether the proportions can be accurately confirmed in your experimental system.
3, If multiple insects are mixed, please add a discussion about the detection sensitivity, which is the percentage above which it can be detected.
4, To make it easier for the reader to understand, you should also add to the introduction which insects are cheap and which insects are sometimes camouflaged instead.
Reviewer 4 Report
This is a very well-written article describing a potential method “DNA metabarcoding” for the identification of insects in foods. The study is properly executed with possible controls and the discussion of the results and draw conclusions is comprehensive and informative. This work should attract high attention, especially in the industry for testing the insects using this novel approach as it not only identifies the insects in foods but also differentiates insect DNA. The manuscript is written clearly which makes this article enjoyable to read and the scientific discussion easy to follow. I solely have only a few suggestions that could potentially help the Authors to further polish this appreciable paper.
1. Typo error abstract line 15, “foodB”.
2. The resolution of figure 1 must be improved for the visibility of readers also it could be put in a landscape format (vertically).
3. In Table 5, the details of the product must be included like the brand, package, and percentage of labeled insect (where available).
4. Referencing must be in the journal's format, for example, year of the publication must be bold instead in parenthesis also keep it along with the volume and page no.
Round 2
Reviewer 1 Report
All changes provided are appropriate and the manuscript is publishable in the present form.
Reviewer 3 Report
I intended to comment so that the paper would be more meaningful to the readers, but I do not feel that the comments I pointed out were properly answered. I have no further comments.